# Self-Stigmatization of Healthcare Workers in Intensive Care, Acute, and Emergency Medicine

**DOI:** 10.3390/ijerph192114038

**Published:** 2022-10-28

**Authors:** Maike Riegel, Victoria Klemm, Stefan Bushuven, Reinhard Strametz

**Affiliations:** 1Institut für Mensch, Arbeit und Psychologie (IMAP), Rennenbergstraße 3, 53639 Königswinter, Germany; 2Wiesbaden Institute for Healthcare Economics and Patient Safety (WiHelP), Wiesbaden Business School, RheinMain UAS, Bleichstr. 44, 65183 Wiesbaden, Germany; 3Institute for Infection Control and Infection Prevention, Hegau-Jugendwerk Gailingen, Health Care Association District of Constance, 78224 Singen, Germany; 4Institute of Medical Education, University Hospital, LMU Munich, 80336 Munich, Germany; 5Department of Anesthesiology and Critical Care, Medical Center-University of Freiburg, Faculty of Medicine, University of Freiburg, 79085 Freiburg im Breisgau, Germany

**Keywords:** self-stigmatization, quantitative study, German hospitals, intensive care, acute medicine, emergency medicine, emotional burden, mental stress

## Abstract

This quantitative study examines whether employees in the fields of intensive care or acute and emergency medicine experience psychological distress because of their daily work. In addition, it was examined if self-stigmatization tendencies can significantly influence the willingness to seek help, and therefore psychological problems are not being treated adequately. These problems lead to various difficulties in professional and private contexts and ultimately endanger patient safety. From May to June 2021, an online questionnaire survey was conducted. This questionnaire combined two validated measuring instruments (PHQ-D and SSDS). To ensure high participation, the departments of anesthesia and/or intensive care medicine in 68 German hospitals were contacted, of which 5 responded positively. A total of 244 people participated in the questionnaire survey. On average, depressive symptoms were of mild severity. At the same time, self-stigmatization regarding depressive symptoms was high. These results highlight the practical need to prepare staff who work in the field of intensive care or acute and emergency medicine at the early onset for potentially traumatic and emotionally demanding events during their university education or studies. Adequate, evaluated, and continuously available support services from the psychosocial field should become an integral part of every staff care structure.

## 1. Introduction

International studies show that healthcare workers are exposed to more emotionally, psychologically, and physically stressful situations in their professional environment than the general population [1,2,3]. Especially in the fields of in-hospital intensive care or acute and emergency medicine (hereinafter referred to as IAN medicine), witnessing potentially traumatic events is part of everyday working life [4]. Therefore, the risk of suffering burnout or post-traumatic stress disorder during one’s professional life is high. Being confronted with suffering, accidents, and death daily carries a high risk of traumatization. Over 90% of the medical and non-medical staff in the rescue service report assignments bearing the risk of traumatization. In addition, at the intra-hospital level, the focus is on extreme professional events that are associated with experiences of violence and threats [5]. Healthcare workers often exceed their own physical, psychological, and emotional-moral stress limits, which in turn impacts the patients [6]. In IAN medicine especially, healthcare workers, such as (specialized) nurses, doctors, and paramedics, are affected. The main causes are an excessive workload, the shrinking scope for action, morally conflicting situations, and dissonance within the team. In addition, due to the political framework conditions, the pressure on healthcare workers is successively increasing, adding to them exceeding their limits [5,7]. Another issue, which is inherent in the described problems, must not be overlooked: the stigma attached to mental illness and the willingness to seek out help for psychological symptoms. Especially in the health sector, the self-sacrificing self- and interpersonal perception dominates, hardly allowing an upcoming need for (psychological) help on the healthcare workers’ part. This leads to the conclusion that psychosocial support services in IAN medicine should be designed to be as low-threshold as possible. In this context, more and more in-hospital care structures and support services for crisis situations have been established in recent years. For example, an increasing number of hospitals have established crisis intervention teams. These are usually made up of trained volunteers and hospital staff and are available to relatives, staff, and, if necessary, patients. However, these services differ greatly in terms of content, quality, and structure. Moreover, they cannot replace primary preventive psychological support. Studies have shown [5,6,8] that staff members see the need for psychological support services but that there is currently a clear discrepancy between needs and existing structures. The following study and its research questions were designed to answer the following hypotheses:There is no relevant psychological stress among healthcare workers;Regarding psychological stress, there are self-stigmatization tendencies among healthcare workers;Healthcare workers are inhibited in their willingness to seek psychological help.

## 2. Materials and Methods

The questionnaire used for the online survey essentially consists of a combination of the questionnaire instruments “Patient Health Questionnaire” (PHQ) as Appendix A and the “Self-Stigma of Depression Scale” (SSDS). All data are available on request.

To detect a mental disorder, the PHQ developed in the USA was used in the German version, the “Gesundheitsfragebogen für Patienten” (PHQ-D). The short form of the PHQ-D was used, which corresponds only to the first page of the full questionnaire and is suitable for assessing depressive disorders, panic disorders, somatoform disorders, anxiety disorders, eating disorders, psychosocial stressors, and psychosocial functioning. The PHQ-D was developed as a viable screening tool in primary care to improve primary healthcare for mental disorders. As it originates from the American language, its diagnostic criteria correspond to those of the DSM-IV. Since the PHQ-D directly questions the diagnostic criteria of the DSM-IV, it has high content validity. In Germany, as well as in most other European countries, the ICD-10 is preferably used as a diagnostic code, so the PHQ-D manual contains a table in which the diagnostic criteria according to DSM-IV and ICD-10 are listed. For the two continuous scales of depressiveness and somatization of the PHQ-D, the calculation of internal consistency is possible based on a categorical evaluation. The internal consistency according to Cronbach’s α is α = 0.88 for the depressiveness scale and α = 0.79 for the somatization scale. The Pearson product-moment correlation of the two scales showed a positive correlation of r = 0.60 (*p* < *0*.001). The two scales can also be evaluated dimensionally so that a determination of the severity of the symptomatology is possible. To evaluate hypothesis 1, the depression scale of the PHQ, the scale sum value was calculated using the nine items of the depression module. The answers gave the participants 0–3 points per item, depending on their answers. For the two continuous scales of depressiveness and somatization of the PHQ-D, the calculation of internal consistency is possible based on a categorical evaluation [9].

The German version of the Self-Stigma of Depression Scale (SSDS) was selected to measure possibly existing stigmatization tendencies. The SSDS is an instrument that measures the extent of self-stigmatizing attitudes of persons with depressive symptoms against themselves. Perceived public or structural stigma is deliberately not included. The wording of the 16 items is kept in the subjunctive; thus, the questionnaire offers the option to ask people regardless of personal experience. The assumption is that the respondents automatically draw on their own experiences that they have had in the context of depression. The aim is to map both the anticipated self-stigma of persons who are not acutely affected and the self-stigma of persons with depression. Internal consistency for the entire SSDS was 0.87 according to Cronbach’s α; for the subscales, it was α = 0.83 for shame, α = 0.78 for self-blame, α = 0.79 for help-seeking inhibition, and α = 0.79 for social inadequacy; test–retest reliability was moderate [10].

The applicability, as well as the psychometric characteristics of a German version of the SSDS, was evaluated by a group of researchers led by Makowski [11] from the Institute of Medical Sociology at the University Medical Centre Hamburg Eppendorf. An examination of construct reliability is still pending. To confirm hypothesis 2, the SSDS was evaluated. To achieve a valid answer, it was evaluated in two steps. At first, the four items related to the dimension of social inadequacy were evaluated. Then, since the German version of the SSDS had no scoring system for the degree of self-stigmatization, the four items were appointed scores. The sum of those scores, similar to the procedure with the PHQ-D, was then used to scale the extent of self-stigmatization.

The version of the SSDS translated in the study is so far the only instrument for surveying depression-specific self-stigma in German-speaking countries. According to Makowski et al., despite methodological limitations, the SSDS represents the first possibility to survey the extent of existing or anticipated self-stigma. Permission to use the instrument was obtained in writing from Dr. Makowski.

To answer the questionnaire, a time-limited link and a QR code were created using an online platform for developing and conducting surveys. Both the link and the QR code could only be used once per end device to ensure no double answers by one person.

The departments of anesthesiology and intensive care medicine, as well as the nursing directorate or the nursing service management of 68 German hospitals (out of 1903 hospitals in Germany in total as of 2020 [12]), were contacted. Hospitals across Germany were contacted regardless of their size or specialization. University hospitals, other maximum care hospitals, and standard care hospitals were contacted to avoid selection bias. The survey, next to typical professions in the field of IAN medicine, included the option for the participants to write down their profession if not listed. This made it possible to add students or paramedics to the survey, which otherwise could not have participated.

## 3. Results

A total of 244 people participated in the survey as shown in Table 1. All datasets could be included in the evaluation.

### 3.1. Hypothesis No. 1: There Is No Relevant Psychological Stress among Healthcare Workers

On average, the participants reached a score of 7.68 for the depression module of the PHQ. When divided into their occupational groups and work fields, doctors working in intensive care on average reached 8.16 points, and nurses in intensive care 8.09 points. Doctors not working in intensive care reached 7.39 points on average, and nurses 8.08 points. The occupational group “others” reached 7.97 points on average. The occupational group “others” was not differentiated between working in the ICU and not because only seven in total stated they were working in the ICU. The participants’ average score of 7.68, regardless if they are working in the ICU or not, indicates that they suffer mild depressive symptoms referring to interpretation as shown in Table 2. The participants reached particularly high scores on the two items “difficulty falling asleep or sleeping through the night, increased sleep” and “tiredness or feeling of having no energy”.

Next to psychological symptoms, somatic symptoms were evaluated as well, and a score was calculated on the somatization scale. There, the participants reached a score of 7.63, meaning a mildly pronounced symptom intensity or somatization on the somatization scale of the PHQ. In the survey, the four items “back pain”, “headache”, “constipation, nervous bowel, or diarrhoea” and “nausea, flatulence, or digestive complaints” achieved high scores.

Therefore, the first hypothesis, that there is no relevant stress among healthcare workers, can be rejected.

### 3.2. Hypothesis No. 2: Regarding Psychological Stress, There Are Self-Stigmatization Tendencies among Healthcare Workers

The lower the score, the less the participants feel socially inadequate, and vice versa. The lowest achievable score per item was 1; the highest was 4.

These average scores in Table 3 show that the participants feel socially inadequate on a higher level.

### 3.3. Hypothesis No. 3: Healthcare Workers Are Inhibited in Their Willingness to Seek Psychological Help

On the help-seeking inhibition subscale, the following items were surveyed:I would be embarrassed to seek professional help for my depression;I would be embarrassed if others knew I was seeking professional help for my depression;If I could not cope, I would not want others to know;I would think I was weak if I took antidepressants.

Analogical to the results on the social inadequacy subscale, the answers were scored, where the lowest possible score was 1, and the highest was 4.

These results shown in Table 4 indicated that healthcare workers are inhibited from seeking psychological help.

## 4. Discussion

The main findings are in accordance with previous studies: the survey’s result shows at least hints of mild depressive disorders among the participants. These are more pronounced in physicians working in the ICU. Regarding the second hypothesis, the study shows that self-stigmatization tendencies exist—also strongly apparent among the physician subgroup. Similar results were found for the third hypothesis: inhibition for seeking psychological help is greatest among physicians working in the ICU.

This study is, as far as known to the authors, the first that describes the phenomenon of self-stigmatization in the context of working in IAN medicine.

Working in the field of IAN medicine generally carries a risk of developing depressive symptoms. The results found in this study correspond with previous studies both from German-speaking countries [5,6,13,14] and the US [7,15,16]. The discovered degree of somatization in this study corresponds to the surveyed psychological distress of the participants. During the research, no comparative studies were found that surveyed a potentially existing somatization explicitly in addition to the psychological stress; therefore, it is advised that more research is carried out on that topic. 

Based on the results of the survey, it is questionable as to what extent self-stigmatization tendencies exist among healthcare workers in IAN medicine regarding depressive symptoms. The tendency to subordinate oneself to stigma regarding psychological issues is overall high, while doctors seem to have stronger tendencies to stigmatize themselves compared to the other occupational groups. This was also found as a result of a study by Rüsch, Angermeyer, and Corrigan [17]. The question arises whether this is related to increasing levels of responsibility or higher levels in the hierarchy and whether the self-image of the healthcare workers triggers self-stigmatization tendencies concerning mental illness.

The effects of mental overload or burnout can be serious for the individual suffering it, the team, and the quality of treatment [18,19,20]. A possible coping mechanism is substance abuse [21]. This does not only impact the healthcare worker negatively but also reduces the quality of care, which further highlights the importance of this subject. Additionally, initial studies [22,23] indicate that the extent of self-stigmatization has a negative impact on the response to therapy and, thus, on the recovery from mental health problems.

This also interacts with the observation that doctors, in particular, achieved similarly high values for the subscale “Inhibitions in Seeking Help”. The one-factor analysis of variance between the three occupational groups, which was calculated for the entire scale of the SSDS, indicates a significant difference in the expression of self-stigmatization tendencies. The Cohen’s d effect strength, which was calculated for the one-factor analysis of variance between the occupational groups “doctors” and “others”, also shows that there are strong differences between these two occupational groups in the expression of the tendency of self-stigmatization. Since the mean values used for the calculation are higher for the occupational group “doctors” than for the group “others”, it can be assumed that doctors tend to stigmatize themselves significantly more than members of the occupational group “others”. For Cohen’s d, the analysis yielded a value of d = 0.8, which indicates a large effect. Specifically, item 7, “If I can’t cope, I wouldn’t want others to know”, scores high. On the other hand, the willingness to seek professional help is high for 64% of the participants, especially if unknown to others. Concerns about taking antidepressants if personally affected are high among 33% of the participants; 67% of the participants would take antidepressants if necessary. No significant differences were found between the occupational group “nurses” and “doctors” or “others” for the total scale of the SSDS.

The results for hypotheses 1–3 illustrate the urgency of eliminating the discrepancy between need and supply concerning psychological support services for healthcare workers in IAN medicine. They also illustrate the need to make those services as low-threshold as possible. That way, the acceptance and, thus, the willingness to participate could be increased. Simultaneously, the employers would fulfill their statutory obligation to implement measures to improve the health and safety of their employees. Even after an intense study of the literature, there were no comparable studies that examined both depressive symptomatology and self-stigmatization tendencies in a survey among healthcare workers, especially in IAN medicine. This could be due to the topic being neuralgic and overall sensitive. Additionally, it might be difficult to collect data on this topic without being able to offer a follow-up concept for the found results. This makes it unattractive for employers to conduct research in their institutions. However, to avoid burnout among many doctors and other healthcare workers, there is a strong need for action on the part of politicians and employers.

The results also show the practical need to prepare healthcare workers in the field of IAN medicine for potentially traumatic and, on an emotional level, exceptionally demanding events, not only during their working careers but also before that in training or during their studies. In order to assure the sustainability of these programs, support strategies have to be inclusive of other fields of support, such as addiction treatment, which could possibly be caused by self-stigmatization.

To be able to generalize this study’s findings, more studies are necessary on a larger scale and more widespread throughout the German healthcare system.

There are limitations to this study. The collected data stem from an online survey, which could distort the information’s validity. The response rate was 7.35%, and only 5 of the 68 contacted hospitals responded positively. This low response rate may be explained by the high workload of hospitals and the high number of survey requests received daily. The study was not conducted by an organization or an official administrative body, possibly lowering the willingness to participate even further. Not all participants that categorized their occupational group as “other” specified their profession, meaning there is no way of telling which occupational groups were part of the study apart from doctors, nurses, MTAs, and some paramedics, which specified their profession with a comment. The response rate of nurses was very low in relation to the other two occupational groups (doctors and others). This problem can also be observed in other studies [6,8]. The nomenclature in the survey could be the reason: not all the nurses working in IAN medicine are registered nurses, thus classifying themselves as “other”. Another reason for the low response rate of nurses could be the lack of computer workspaces and thus being inhibited from filling out the survey. In a further step, the questionnaire should be tested on a comparative population to differentiate whether answers from healthcare workers in IAN medicine differ significantly from the responses of other professional groups. Additionally, due to the survey being conducted during the COVID-19 pandemic with its restrictions (e.g., social distancing), the scores might have been higher than if the survey had been conducted before the pandemic. However, studies investigating the Second Victim Phenomenon, which may be associated with these symptoms, do not show significant deviation before [24] and during [25] the pandemic. We therefore acknowledge a possible influence of COVID-19 but do not assume a significant confounding of our results due to the pandemic.

## 5. Conclusions

Adequate, evaluated, and continuously available support services from the psychosocial field should become an integral part of every staff care structure to process traumatic and emotionally challenging events. To be able to provide and evaluate these care structures, the interdisciplinary cooperation between research, practice, and the respective professional groups should be strengthened to develop and ideally implement effective programs for the prevention of depressive symptoms, burnout, post-traumatic reactions, etc. These programs must cover a range of topics related to the topic “psycho-social care.” These must include a thematic complex on the concept of stigma, especially self-stigma. This would maintain and/or increase the resilience, well-being, and performance of healthcare workers, and the employers would fulfill their statutory duty of care towards their employees, which in turn could lead to better employee retention and the overall reputation of healthcare institutions. However, this is very likely only possible in the accompanying context of political changes. Politicians should advocate for an orderly structural change with a concordant definition of goals at the federal and state levels to counteract a culture of mistrust and controlling over-bureaucracy.

To ensure a well-functioning healthcare system, it is necessary to empower healthcare workers and take their emotional and psychological problems seriously; therefore, measurements need to be taken to strengthen their resilience and to give them professional psychological help if needed. Additionally, it is necessary to implement a speak-up culture that does not stigmatize psychological problems, so those affected can speak openly about them and can overcome them in an accepting environment.

## Figures and Tables

**Table 1 ijerph-19-14038-t001:** Characteristics of the participants.

Characteristics		Number of Participants
	Female	119
Gender	Male	124
	Non-specified	1
	20–30	20
	30–40	58
Age Groups (Doctors)	40–50	35
	50–60	14
	60–70	6
	20–30	2
	30–40	7
Age Groups (Nurses)	40–50	11
	50–60	11
	60–70	1
	20–30	43
	30–40	18
Age Groups (Other)	40–50	7
	50–60	8
	60–70	0
	Doctors	133
	Nurses	32
Professions	Medical Technical Assistants	2
	“Other”	76
	Non-specified	1
Work Field	Intensive Care Unit (ICU)	118
	Non-ICU	126

**Table 2 ijerph-19-14038-t002:** Scores and their interpretations.

Score	Interpretation
0–4 points	No depressive disorder
5–9 points	Mild or subthreshold depressive disorder
10–14 points	Moderate depressive disorder
15–19 points	Moderately severe depressive disorder
20–27 points	Severe depressive disorder

**Table 3 ijerph-19-14038-t003:** Average score on the social inadequacy subscale.

Occupational Group	Average Score
Doctors	2.76
Nurses (non-ICU)	2.49
Nurses (ICU)	2.41
Others	2.21

**Table 4 ijerph-19-14038-t004:** Average scores on the help-seeking inhibition subscale.

Occupational Group	Average Score
Doctors (non-ICU)	2.41
Doctors (ICU)	2.45
Nurses (non-ICU)	2.27
Nurses (ICU)	2.23
Others	2.24

## Data Availability

The data presented in this study are available on request from the corresponding author.

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
