# Peer review of "Self-Stigmatization of Healthcare Workers in Intensive Care, Acute, and Emergency Medicine"

_ijerph, 2022, doi:10.3390/ijerph192114038_

Round 1
Reviewer 1 Report
Your article represents yet another alarm signal for health care professionals, who are daily subjected to stressful situations and negative events, from an emotional point of view. Please provide detailed answers to the following questions (observations):
1. How do you explain the small number of hospitals participating in the study proposed by you?
2. On the occasion of the periodic medical check-up carried out by the occupational medicine service, is psychological testing applied? How often? Is there feedback at this level related to the stress or burnout at the workplace of the medical staff?
3. From my point of view, although I understand the interference of the political side in the medical field, nevertheless, I suggest that in the "Discussions" section to avoid the presentation related to the DRG, I do not consider it so relevant for the article.
Author Response
Reviewer #1
Your article represents yet another alarm signal for health care professionals, who are daily subjected to stressful situations and negative events, from an emotional point of view. Please provide detailed answers to the following questions (observations):
- How do you explain the small number of hospitals participating in the study proposed by you?
Thank you for your absolutely justified comment, we have explained the low response rate in detail in the discussion section as followed:
This low response rate may be explained with the high work load of hospitals and the high number of survey requests received daily. The study wasn’t conducted by an organization or an official administrative body, possibly lowering the willingness to participate even further
- On the occasion of the periodic medical check-up carried out by the occupational medicine service, is psychological testing applied? How often? Is there feedback at this level related to the stress or burnout at the workplace of the medical staff?
Thank you for this important question. You are absolutely right about periodic check-ups being carried out by occupational medicine service. In Germany, there is the so called G42 check-up which addresses the danger of infections for medical staff. This check-up focusses on the prevention of infection with Hepatitis-B, -C and HIV. To our knowledge, check-ups of psychologic health are neither mandatory nor regularly applied by occupational medicine services in Germany.
- From my point of view, although I understand the interference of the political side in the medical field, nevertheless, I suggest that in the "Discussions" section to avoid the presentation related to the DRG, I do not consider it so relevant for the article.
We absolutely agree with your point that highlighting DRGs may not be of any interest for an international audience. We therefore removed this section completely.
Reviewer 2 Report
I reviewed "Self-Stigmatization of Health Care Workers in Intensive Care, Acute, and Emergency Medicine" by Riegel et al., a manuscript preseting results from an internet-based study of psychological distress among health care workers in emergency units (as the title obviously states). The language used in the study is understandable, the concept is interesting and important because often healthcare workers feel invincible and this is wrong as - over the time - this situation leads to depression and psychoactive substances abuse. The design and methodology are well thought and adequately prepared. The discussion covers all the aspects it should cover. The manuscript, in my humble opinion, doesn't need much work before being published, but I am a psychiatrist so I would like to give you three hints (also editorial ones) to make the manuscript even better.
1. In the abstact you state "depression was of mild severity". You can't tell if someone is depressed just from a self-report questionnaire, as in accordance to DSM-V and ICD-10/ICD-11, one has to be examined personally by a physician. Instead of "depression", you shold use "depressive symptoms" or "depressive items on the scale", as you wish.
2. COVID-19 restrictions were terrible for our mental health in general. In my opinion, putting an information in the limitations that would highlight the fact of gathering data during COVID-19 restrictions would be useful, as the scores might've been a little higher (or not and that we won't know now) than the usual in your population. This doesn't make the study less valuable but it puts more transparency. I would also list the limitations in the end of discussion as it usually is listed.
3. Consider highlighting in the discussion how stress puts healthcare worker at risk of substance abuse and how current healthcare is not enough. For example - https://journals.sagepub.com/doi/full/10.1177/00333549211058176 . From my personal work experience (ask any psychiatrist as sadly it's a European trend) is that emergency care workers are at higher risk of substance abuse because of easy access to opioids and anesthetics. You don't have to list that but highlighting how efforts to provide proper mental health care which includes addictions treatment for such workers is crucial for making emergency care sustainable would be great, as this is directly connected to the subject of your research.
Best Regards
Reviewer (MD, psychiatrist)
Author Response
Reviewer #2
I reviewed "Self-Stigmatization of Health Care Workers in Intensive Care, Acute, and Emergency Medicine" by Riegel et al., a manuscript preseting results from an internet-based study of psychological distress among health care workers in emergency units (as the title obviously states). The language used in the study is understandable, the concept is interesting and important because often healthcare workers feel invincible and this is wrong as - over the time - this situation leads to depression and psychoactive substances abuse. The design and methodology are well thought and adequately prepared. The discussion covers all the aspects it should cover. The manuscript, in my humble opinion, doesn't need much work before being published, but I am a psychiatrist so I would like to give you three hints (also editorial ones) to make the manuscript even better.
Thank you very much for your time to review our paper. We are humbled by your positive feedback and would like to comment on every aspect of your very valuable suggestions.
- In the abstract you state "depression was of mild severity". You can't tell if someone is depressed just from a self-report questionnaire, as in accordance to DSM-V and ICD-10/ICD-11, one has to be examined personally by a physician. Instead of "depression", you shold use "depressive symptoms" or "depressive items on the scale", as you wish.
Thank you very much, it is indeed not possible to detect depression from a self-report questionnaire. We therefore changed it according to your suggestion to “depressive symptoms” throughout the article.
- COVID-19 restrictions were terrible for our mental health in general. In my opinion, putting an information in the limitations that would highlight the fact of gathering data during COVID-19 restrictions would be useful, as the scores might've been a little higher (or not and that we won't know now) than the usual in your population. This doesn't make the study less valuable but it puts more transparency. I would also list the limitations in the end of discussion as it usually is listed.
We absolutely agree that COVID 19 may have influenced our results, but studies conducted during COVID-19 on the Second Victim Phenomenon indicate that COVID-19 itself had only little impact on self-reported symptoms. We have added the following paragraph to address this important consideration:
Also, due to the survey being conducted during the COVID-19 pandemic with its restrictions (e.g., social distancing), the scores might have been higher than if the survey was conducted before the pandemic. However, studies investigating the Second Victim Phenomenon which may be associated with these symptoms do not show significant deviation before [25] and during [26] the pandemic. We therefore acknowledge a possible influence of COVID-19, but do not assume a significant confounding of our results due to the pandemic.
- Consider highlighting in the discussion how stress puts healthcare worker at risk of substance abuse and how current healthcare is not enough. For example - https://journals.sagepub.com/doi/full/10.1177/00333549211058176 . From my personal work experience (ask any psychiatrist as sadly it's a European trend) is that emergency care workers are at higher risk of substance abuse because of easy access to opioids and anesthetics. You don't have to list that but highlighting how efforts to provide proper mental health care which includes addictions treatment for such workers is crucial for making emergency care sustainable would be great, as this is directly connected to the subject of your research.
Thank you very much, we have added the following paragraph and cited the article you suggested:
A possible coping mechanism is substance abuse [21]. This doesn’t only impact the health care worker negatively, but also reduces the quality of care which further highlights the importance of this subject.
We also discussed sustainability of interventions and added the following paragraph:
In order to assure sustainability of these programs, support strategies have to be inclusive of other fields of support like addiction treatment, which could possibly be caused by self-stigmatization.
Best Regards
Reviewer (MD, psychiatrist)